# Targeted randomization dose optimization trials enable fractional dosing of scarce drugs

**Philip S. Boonstra[1,2], Alex Tabarrok[3], Garth W. Strohbehn**[2,4,5,6,7] *

**1** Department of Biostatistics, University of Michigan, Ann Arbor, Michigan, United States of America,
**2** Rogel Cancer Center, University of Michigan, Ann Arbor, Michigan, United States of America,
**3** Department of Economics, George Mason University, Fairfax, Virginia, United States of America,
**4** Veterans Affairs Center for Clinical Management and Research, Ann Arbor, Michigan, United States of America, **5** Institute for Health Policy and Innovation, University of Michigan, Ann Arbor, Michigan, United States of America, **6** Division of Medical Oncology, LTC Charles S Kettles VA Medical Center, Ann Arbor, Michigan, United States of America, **7** Division of Hematology/Oncology, Department of Internal Medicine, University of Michigan, Ann Arbor, Michigan, United States of America

* gstrohbe@umich.edu

**Data Availability Statement:** The code used in simulated clinical trials is available as an R package on the first author's GitHub page: https://github.com/psboonstra/DoseDeescalation, or installed

## Abstract

Administering drug at a dose lower than that used in pivotal clinical trials, known as fractional dosing, can stretch scarce resources. Implementing fractional dosing with confidence requires understanding a drug's dose-response relationship. Clinical trials aimed at describing dose-response in scarce, efficacious drugs risk underdosing, leading dose-finding trials to not be pursued despite their obvious potential benefit. We developed a new set of response-adaptive randomized dose-finding trials and demonstrate, in a series of simulated trials across diverse dose-response curves, these designs' efficiency in identifying the minimum dose that achieves satisfactory efficacy. Compared to conventional designs, these trials have higher probabilities of identifying lower doses while reducing the risks of both population- and subject-level underdosing. We strongly recommend that, upon demonstration of a drug's efficacy, pandemic drug development swiftly proceeds with response-adaptive dose-finding trials. This unified strategy ensures that scarce effective drugs produce maximum social benefits.

## Introduction

Shortages of supplies, the labor and matériel that enable safe delivery of care, and the means of paying for that care generate marked disparities in global health outcomes [1]. COVID-19 has only exacerbated disparities resulting from scarcity [2]. Though increased production and distribution of scarce goods are often pursued, these tactics are insufficient to address exponentially escalating global pandemics where demand for a scarce resource quickly outstrips its supply. Under scarcity, efficient, ethical rationing according to ethical principles, such as benefit maximization, inequity minimization, or care of the worst off, becomes necessary [1].

A complementary approach, dose fractionation, rations the amount of a divisible scarce resource that is allocated to each individual recipient [3–6]. Fractionation is a utilitarian attempt to produce "the greatest good for the greatest number" by increasing the number of recipients who can gain access to a scarce resource by reducing the amount that each person receives,

directly from R using devtools::install_github ("psboonstra/DoseDeescalation").

**Funding:** PSB and GWS are supported in this work by National Cancer Institute (US), grant P30CA046592. The funders had no role in study design, data collection and analysis, decision to publish, or preparation of the manuscript.

**Competing interests:** PSB has received research funding from Bristol Myers Squibb and Janssen outside of the submitted work. AT has no conflicts to disclose. GWS serves as an uncompensated Director of the Optimal Cancer Care Alliance. GWS is a co-inventor of a patent held by the University of Chicago covering the use of low-dose tocilizumab in the treatment of viral infections. GWS reports no material conflicts of interest with regards to contract research organizations, biostatistical firms, or vaccines. GWS is employed by the United States Department of Veterans Affairs; this work does not represent the official position of the United States Department of Veterans Affairs. There are no patents, products in development or marketed products to declare. This does not alter our adherence to PLOS ONE policies on sharing data and materials.

acknowledging that individuals who receive lower doses may be worse off than they would be had they received the "full" dose. If, for example, an effective intervention is so scarce that the vast majority of the population lacks access, then halving the dose in order to double the number of treated individuals can be socially valuable, provided the effectiveness of the treatment falls by less than half. For variable motivations, vaccine dose fractionation has previously been explored in diverse contexts, including Yellow Fever, tuberculosis, influenza, and, most recently, monkeypox [7–12]. Modeling studies strongly suggest that vaccine dose fractionation strategies, had they been implemented, would have meaningfully reduced COVID-19 infections and deaths [13], and perhaps limited the emergence of downstream SARS-CoV-2 variants [6]. The fractionation strategy is both ethically and economically dominant as long as the dose used is equal to or greater than the minimum dose that produces satisfactory, near-maximal efficacy (MDSE) [14].

Confident employment of fractionation requires knowledge of a drug's dose-response relationship [6, 13], but direct observation of both that relationship and MDSE, rather than pharmacokinetic modeling, appears necessary for regulatory and public health authorities to adopt fractionation [15, 16]. Oftentimes, however, early-phase trials of a drug develop only coarse and limited dose-response information, either intentionally or unintentionally. A speed-focused approach to drug development, which is common for at least two reasons, tends to preclude dose-response studies. The first reason is a strong financial incentive to be "first to market." The majority of marketed cancer drugs, for example, have never been subjected to randomized, dose-ranging studies [17, 18]. The absence of dose optimization may raise patients' risk. Further, in an industry sponsored study, there is a clear incentive to test the maximum tolerated dose (MTD) in order to observe a treatment effect, if one exists. The second reason, observed during the COVID-19 pandemic, is a focus on speed for public health. Due to ethical and logistical challenges, previously developed methods to estimate dose-response and MDSE have not routinely been pursued during COVID-19 [19]. The primary motivation of COVID-19 clinical trial infrastructure has been to identify *any* drug with *any* efficacy rather than maximize the benefits that can be generated from each individual drug [3, 18, 20, 21]. Conditional upon a therapy already having demonstrated efficacy, there is limited desire on the part of firms, funders, or participants to possibly be exposed to suboptimal dosages of an efficacious drug, even if the lower dose meaningfully reduced risk or extended benefits [16]. Taken together, then, post-marketing dose optimization is a commonly encountered, high-stakes problem–the best approach for which is unknown.

Both health equity- and supply-minded investigators who seek to maximize the benefits that can be achieved from a fixed supply of an efficacious drug may be better able to achieve their goals using adaptive approaches. Such a strategy would seek MDSE while limiting the potential harms to individual trial subjects and society that might result from administration of lower doses. With that motivation, we present in this manuscript the development an efficient trial design and treatment arm allocation strategy that quickly de-escalates the dose of a drug that is known to be efficacious to a dose that more efficiently expands societal benefits. For the purposes of simplification, we target 80% relative efficacy, but the methods developed here generalize to other relative efficacy targets. We intend for these methods to be used as the second step in a two-step approach to pandemic drug development aimed at first proving efficacy and second maximizing benefits [4].

## Methods

### Model formulation

We assume that there are T dose levels to be assessed, labeled from 1 to T. Dose-level 0 (placebo) will not be assessed, since we assume in this scenario that the medication under study

has already been demonstrated efficacious against a placebo control in a previous trial. Response, $R$, is the primary (dichotomous) outcome of the trial. Dose-level and $R$ are linked by an unknown dose-response function:

$$\xi_t \equiv \Pr(R = 1|t), t = 1, \ldots, T$$

$\xi_t$ is the probability of response when dose-level t is used. Assuming that response is conditionally independent across subjects given dose-level t, R is thus distributed as a Bernoulli random variable with unknown parameter $\xi_t$. Standard approaches for estimating $\xi_t$, such as separate maximum likelihood estimates of each $\xi_t$, would not impose any constraint on the relationship between estimates of $\xi_t$ and $\xi_{t-1}$. One mild assumption that is often sensible is that $\xi_{t-1} \leq \xi_t$; that is, that the true dose-response curve is non-decreasing. Imposing this assumption would improve statistical efficiency and allow for smaller, faster trials without sacrificing performance, provided the assumption is correct. Notably, *improved* efficacy at lower doses (e.g., tuberculosis vaccine H56+IC31 [12]) is not modeled but is allowed.

Monotonicity can be achieved by pre-specifying a parametric dose-response model (e.g., logistic regression) [22]. Alternatively, one can avoid pre-specifying the functional relationships between each $\xi_t$ by equipping a non-parametric model with a Bayesian prior that enforces monotonicity. Our prior makes use of the horseshoe isotonic probability vector distribution [23]. Briefly, we define a length T+1 set of non-negative parameters (weights)–$\alpha_1$, $\alpha_2$, ..., $\alpha_T$, $\alpha_{T+1}$ –that correspond one-to-one to the response probabilities for each dose-level according to the equation:

$$\xi_t = \sum_{j=1,\ldots,t} \alpha_j / \sum_{j=1,\ldots,T+1} \alpha_j$$

That is, the response probability $\xi_t$ is a function of the sum of the first t weights divided by the total sum of the weights. Since the weights are non-negative, this transformation ensures that response probabilities will be non-decreasing. The prior placed on each $\alpha_t$ is the positive half of the horseshoe (HS) or the horseshoe-plus (HSPl) distribution, which are described in the Supplemental methods in S1 Appendix [24–29]. With the HS, values of $\alpha_t$ are allowed to be quite close to zero, representing portions of dose-response curves in which the change in dose-level produces little-to-no increase in response probability, or quite large, representing portions of dose-response curves in which change in dose-level markedly increases response probability. The HSPl sharpens this distinction between small and large increases even further. Both prior formulations enable the flexible estimation of monotonic dose-response curves without imposing constraints on the shape of the curve.

A Bayesian analysis calculates the posterior distribution of the parameters. The fitted posterior, straightforwardly calculated in the Stan programming language [30, 31], is proportional to the product of the prior and the model's likelihood. As described in the next section, this posterior distribution is calculated during the trial to make dosing assignments and at the end of the trial to conduct a final inference. A computationally efficient implementation of this Bayesian calculation is provided in the Supplemental methods in S1 Appendix.

## Dose-level assignment algorithms

Based upon this Bayesian model formulation, we proposed algorithms that will de-escalate dose levels between trial subjects, from the known dose at level *T* towards the MDSE, defined as a smaller (lower) dose at the level that retains $100k\%$ of dose at level *T*'s efficacy. In this setting, $k$ is a prespecified design parameter that reflects the degree to which the relative efficacy is 'satisfactory' (in MDSE). For the hypothetical purpose of vaccine dose fractionation [13], we

assign $k$ a value of 0.8. Symbolically, MDSE is defined as:

$$\min\{t : \xi_t \geq k \cdot \xi_T\} \tag{1}$$

Generically, a de-escalation design will enroll either an individual subject or cohort of subjects at dose-level $t$, with the first such dose-level denoted $T$ (the highest dose-level in the study). After administering drug at dose-level $t$, response(s) among the subject(s) are recorded. Based on the responses for the current dose-level $t$, each of the dose-levels' posterior probability of being the true MDSE is calculated. Because the prior assumes dose-response to be non-decreasing, information is 'borrowed' across dose-levels (e.g., an unsuccessful outcome for the second-lowest dose-level will tend to decrease the probability of being true MDSE for *both* the second-lowest *and* lowest dose-levels). We denote this posterior probability for dose-level $t$ calculated after the first n patients as $\rho_t(n)$. Using $\rho_t(n)$, the next subject(s) enrolled is/are assigned to a dose-level. Since exactly one dose-level must be the MDSE, the sum of $\rho_t(n)$ across all the dose-levels will always be 1 and the mean $\rho_t(n)$ will always evaluate to 1/T (where T is the number of dose-levels evaluated in the trial). Depending on the dose assignment mechanism, the next subject(s) may be assigned to higher dose-level(s). The trial progresses until a stopping rule is encountered; in the clinical trial simulations, the stopping rule was reaching the pre-specified sample size.

We assessed five algorithms by which dose assignments could reasonably be made, summarized in Table 1. The first, called Naïve, *always* assigns the next subject to the dose-level with highest posterior probability of being the MDSE given the current data (i.e., $\text{argmax}_t\ \rho_t(n)$), in accordance with the trial's objective. Initial investigations into Naïve performance suggested local bias (i.e., getting 'stuck' near the starting dose-level $T$); the other four algorithms were generated to combat this. Greedy (Gr), assigns the next subject to the smallest dose-level $t$ with posterior probability exceeding the mean posterior probability across all dose-levels; that is, Gr assigns $\min\{t: \rho_t(n) > 1/T\}$. Targeted Randomization (TR) identifies the dose-level most likely to be MDSE, $\text{argmax}_t\ \rho_t(n)$ (as in Naïve), but randomizes the next subject(s) to either it or one of the two dose-levels immediately below it. The probability of randomization across three dose-levels explored in TR is proportional to the posterior probability that each is MDSE; if

**Table 1. Comparison of dose level allocation strategies across a range of posterior probabilities.**

|  | Dose Level 1 | Dose Level 2 | Dose Level 3 | Dose Level 4 | Dose Level 5 | Dose Level 6 | Mean P(MDSE) |
|---|---|---|---|---|---|---|---|
| **Estimated P(MDSE)** | 0.1 | 0.2 | 0.4 | 0.2 | 0.09 | 0.01 | $0.166\bar{6}$ |
| **Next Subject's Probability of Allocation** | | | | | | | |
| **Schema** | P(Dose Level 1) | P(Dose Level 2) | P(Dose Level 3) | P(Dose Level 4) | P(Dose Level 5) | P(Dose Level 6) | |
| **EA** | $0.166\bar{6}$ | $0.166\bar{6}$ | $0.166\bar{6}$ | $0.166\bar{6}$ | $0.166\bar{6}$ | $0.166\bar{6}$ | |
| **Naïve** | 0 | 0 | 1 | 0 | 0 | 0 | |
| **Gr** | 0 | 1 | 0 | 0 | 0 | 0 | |
| **TRInd** | 0.1429 | 0.2857 | 0.5714 | 0 | 0 | 0 | |
| **TRCoh\*** | 0.1429 | 0.2857 | 0.5714 | 0 | 0 | 0 | |

In the top half of the table, the estimated probability of each respective dose level being MDSE is given, ranging from 0.01 to 0.4. Mean probability of a given dose level in this setup (6 dose levels) is, definitionally, $1/6 = 0.166\bar{6}$. The second half of the table demonstrates the variation in probability of allocation to a given dose level amongst the different allocation strategies, given the MDSE probability distribution above. Note that the probabilities of TRInd and TRCoh are equal in this example; TRCoh only updates estimated probability of and, consequently, allocation probabilities after a discrete trial cohort has been filled. **Abbreviations:** P(MDSE), probability of a given dose level being MDSE; EA, equal allocation; Naïve, naïve allocation; Gr, greedy allocation; TRInd, targeted randomization, individual; TRCoh, targeted randomization, cohort.

the data suggest that one of the dose-levels is unlikely to be MDSE, then subjects will rarely be randomized to it. We consider two TR variants, depending on the trial's goals and time/cost constraints: TRInd updates the posterior probabilities after each individual subject's outcome and TRCoh updates the posterior probabilities after each 1/3 of the planned sample size is enrolled. TRCoh is expected to be more logistically feasible when accrual is fast relative to the follow-up required to observe response. Finally, Equal Allocation (EA) is a non-adaptive dose assignment mechanism that is insensitive to the posterior probabilities, akin to a randomized dose-ranging study *sans* interim analysis, such that all dose levels receive an equal number of subjects by the end of the trial.

### Dose-response curves

For purposes of generating "true dose-response curve" data for use in simulations, we considered 32 distinct dose-response curves (Fig 1) each with true dose-response probabilities governed by parameterizations of the four-parameter Hill equation [32]:

$$\Pr(R = 1|X) = a + \frac{(b - a)}{\left[1 + \left(\frac{c}{x}\right)^d\right]}$$

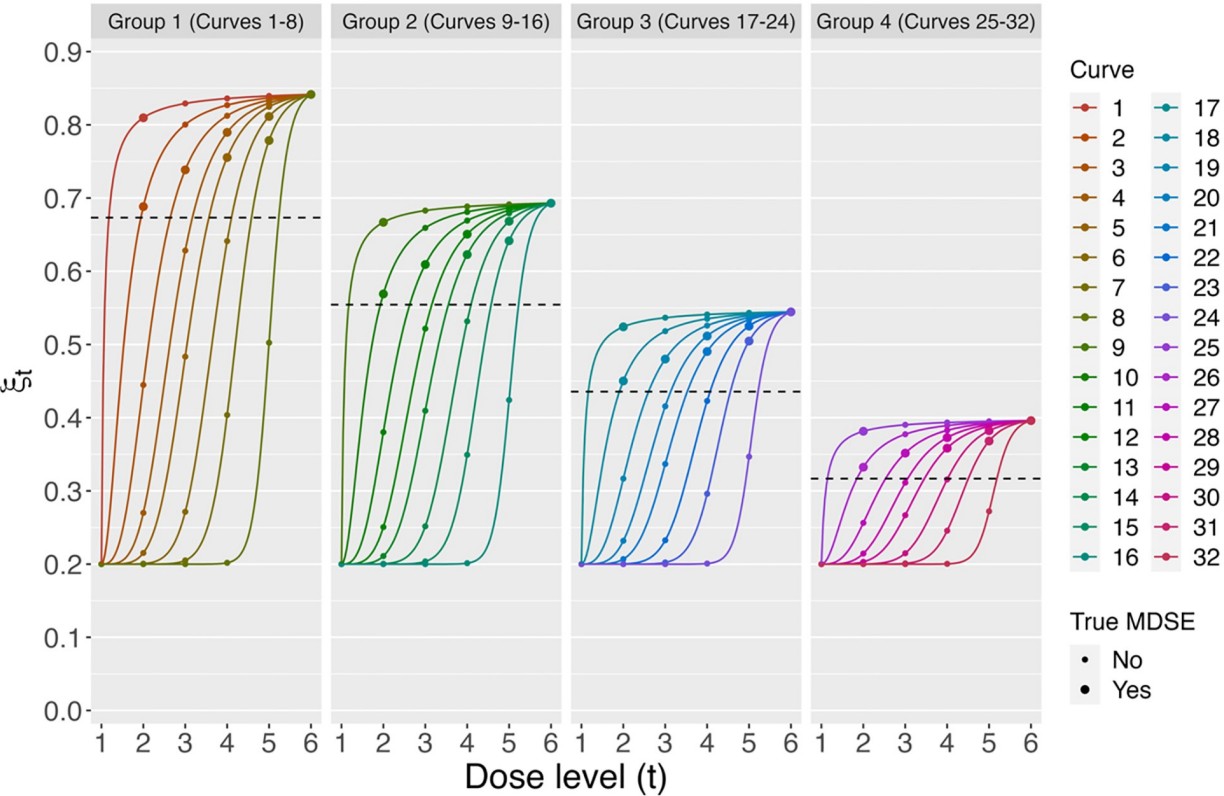

**Fig 1. True dose-response distribution (conditional on dose level) used in simulated clinical trials.** Curves were selected to capture a range of MDSEs and rates at which efficacy declines in response to dose de-escalation. Hill equation parameters of true dose-response curves are summarized in **S1 Table in** S1 Appendix.

The parameter $a$ is the response probability when X = 0 and also the lowest probability of response; $b$ is the upper asymptote and the highest theoretical probability of response; $c$ is the value of X that achieves half of the maximum possible increase in probability, i.e. the average of $a$ and $b$; and d controls the rate at which efficacy increases with increasing X. Parameter values of the 32 true dose-response curves that we considered are summarized in Fig 1 **and S1 Table in** S1 Appendix. Curves 1–8 (each with $a = 0.2$, $b = 0.85$, $c = 0.7$, and $d$ varying from 1 to 20) are considered in the main text, while Curves 9–32 (having smaller values of $b$: 0.70 [Curves 9–16], 0.55 [Curves 16–24], or 0.4 [Curves 25–32]) are evaluated in the Supplement. For each curve, we assumed that T = 6 dose levels are evaluated in a trial and thus selected six corresponding values of X (we explicitly did not include dose levels with zero probability of efficacy). The smallest dose level (labeled dose-level 1) was always at X = 0, thus the true (unknown) value of $\xi_1$ was always equal to $a$. The largest dose level (labeled dose-level $T = 6$) was at the value of X inducing (in truth) 99% of the maximum response probability, i.e. the value of X solving $\Pr(R = 1 | X) = 0.99b$, so that $\xi_T = 0.99b$. The four intermediate dose levels were the values of X equally spaced on the X scale between the two extreme dose levels. Since the largest dose level considered always has true probability of response equal to $0.99b$, the true MDSE in these scenarios as defined in Eq 1 reduces to $\min\{t: \xi_t \geq k \cdot 0.99 \cdot b\}$. **Remark:** We note that the Hill equation was only used as a tool to generate the true values of the response probabilities $\xi_1, \xi_2, \ldots, \xi_6$, whereas the estimation of these probabilities (which are unknown parameters in the trial) is conducted using the isotonic probability vector distributions (**Supplemental Methods in** S1 Appendix).

## Simulated clinical trials and sensitivity analyses

The base design used one of the five dose-assignment algorithms above together with a target relative threshold of $k = 0.8$, the HS prior on the dose-response probabilities, and a total sample size of $n = 102$ (i.e. an average of 17 observations per dose-level). The appropriate sample size for a given trial will be driven by statistical, logistical, and budgetary considerations, and our choice is not meant to be a global recommendation. To quantify how performance changes with sample size, we also considered n = 144 and n = 201, each of which increases the total amount of statistical information in the trial by a factor of about 1.4 relative to the smaller sample size. Previous economic modeling suggested $k = 0.7$ to be a reasonable relative efficacy target for dose fractionation of a pandemic vaccine [13]; we chose $k = 0.8$ for our simulations to provide additional margin of safety. The relative threshold $k$ remained 0.8 for sensitivity analyses, meaning the MDSE was defined as the smallest dose level that achieved absolute efficacy $\geq$ 0.6732 (curves 1–8), 0.5544 (curves 9–16), 0.4356 (curves 17–24), or 0.3168 (curves 25–32). The relative efficacy target for dose fractionation is idiosyncratic to each situation, thus the k used in this study is somewhat arbitrary and not generalizable. The targeted randomization approach concentrates study participants to the dose levels that require the highest resolution. Our dataset includes difficult "edge cases" where a dose level's "true" efficacy is just below the threshold level of acceptable efficacy (e.g., curves 4, 6, 12, 13, 20, 22, 28, and 30). Changing k (i.e., moving the dotted line in Fig 1 higher or lower) would not change whether or not these difficult edge cases exist, it would simply change which of the true dose-response curves the edge cases are. On the whole, we would not expect appreciable changes in the operating characteristics, nor would we expect the relative performances of the 5 dose-level randomization schemes to change. To gauge the sensitivity to choice of prior, we considered the HSPL as an alternative prior formulation. Across all combinations of data generating mechanisms, designs, and primary and sensitivity analyses, 960 distinct clinical trial scenarios were simulated—32 distinct true dose-response curves (curves 1–32), 2 alternative prior

distributions for $\alpha_j$ (HS and HSPL), 5 dose assignment strategies (Naïve, Gr, TRInd, TRCoh, and EA), and 32), and three total sample sizes (n = 102, n = 144, and n = 201)–with $N$ = 500 simulations for each combination.

## Operating characteristics

We evaluated several operating characteristics assessing the societal benefits, societal risks, and individual risks of the different dose allocation strategies (Table 2). The goal of the trial is to identify the true MDSE; we therefore evaluated the likelihood that a dose allocation strategy's estimated MDSE is equal to the true MDSE for a given simulated clinical trial, with higher proportions being better. Under scarcity, estimating the MDSE quickly is also critical. For drugs that have dose-dependent toxicity, subjects are better served, relative to the status quo, by receiving the "correct" dose level: We therefore present the mean proportion of subjects who are assigned to the true MDSE within a given simulated clinical trial.

Societal risk is increased if the *trial's* estimated MDSE is less than the true MDSE, leading to widespread underdosing: We therefore report, for a given simulated clinical trial, the proportion of $N$ = 500 simulations for which the estimated MDSE is *less* than the true MDSE. Societal risk is increased if, in seeking the MDSE, massive error is injected into the point estimates of a drug's maximal efficacy. Under this trial framework, de-escalation does not occur if the estimated absolute response rate of higher dose levels is substantially less than 100$k$%: We therefore present the root-mean-squared error (RMSE) of the estimated response probability at the estimated MDSE. This operating characteristic does not penalize a trial for misestimating MDSE–the purpose of the trial is to estimate MDSE relative to dose level T, whatever it may be. While the individual enrolling in a dose optimization trial may or may not care about the probability of the trial itself identifying the true MDSE, he/she is likely to care about his/her *personal* probability of being assigned to a dose level less than the true MDSE, in which case he/she would be *worse* off for having enrolled in the trial (assuming he/she would otherwise have access to the drug): We therefore assess the probability that an individual trial subject receives a dose lower than the true MDSE.

**Table 2. Summary features of operating characteristics of simulated clinical trials.**

| Operating Characteristic | Estimand | Interpretation | Societal- or Individual-Level Effect | Translation at Scale |
|---|---|---|---|---|
| True MDSE Identification | $P(\widehat{MDSE} = MDSE)$ | Probability of correctly estimating MDSE | Societal | Probability that a de-escalation trial identifies a true MDSE that expands access |
| Risk of MDSE Underestimation | $P(\widehat{MDSE} < MDSE)$ | Probability of estimating MDSE less than the true MDSE | Societal | Risk that a de-escalation trial leads to underdosing at population scale |
| RMSE Efficacy | $\sqrt{E[(\widehat{\hat{\xi}_{MDSE}} - \widehat{\xi_{MDSE}})^2]}$ | Square root of the average squared distance between estimated response at estimated MDSE and true response at estimated MDSE | Societal | Error introduced by the trial to estimates of drug's efficacy |
| Administering a Dose Less than True MDSE | $E[\sum_{t=1}^{MDSE-1} m_{tn}]/n$ | Average number of subjects enrolled at sub-MDSE dose levels | Individual | Risk of underdosing that is assumed by an individual clinical trial participant |

Societal and individual outcomes of interest from simulated clinical trials. At a societal level, de-escalation seeks to: 1) expand drug access and maximize benefits (by correctly identifying true MDSE), while, simultaneously, 2) minimizing the chances that the drug is underdosed at a population level (by avoiding MDSE underestimation) and 3) minimizing the error injected into discourse (by accurately estimating the efficacy of the drug being tested). Societal level benefits also need to be weighed against the risk to individual subjects who would participate in these trials: Trial designs should, where possible, reduce the risk of administering a dose of drug less than true MDSE to subjects.

## Results

### Societal benefit: True MDSE identification

Accuracy in true MDSE identification of the 5 dose allocation strategies in the base case is summarized in Fig 2. EA had peak performance at the extrema–dose-response curves for which true MDSE was either the starting or lowest dose levels, correctly estimating MDSE in 84% of simulated trials. For dose-response curves where true MDSE was 2 to 4 dose levels lower than *T* (e.g., curves 2–6), EA accurately estimated MDSE in 40–60% of simulated trials. Compared to EA, TRCoh and TRInd demonstrated consistently better MDSE estimation for these non-extreme dose-response curves. Naïve demonstrated "stickiness", with poor MDSE estimation for dose-response curves having true MDSE lower than dose level *T* (curves 1–7) (e.g., for curve 7, Naïve 58% vs EA 84% vs TRCoh 92%). Gr demonstrated improved MDSE estimation for curves requiring de-escalation (e.g., for curve 7, Naïve 58% vs Gr 87%). Gr demonstrated poorer MDSE estimation than EA, TRCoh, and TRInd (e.g., for curve 3, Gr 26% vs EA 53% vs TRCoh 59% vs TRInd 49%). TRCoh demonstrated better MDSE estimation than TRInd for dose-response curves where the absolute efficacy of true MDSE was close to threshold (e.g., for curve 2, true MDSE at dose level 2 with efficacy slightly higher than 0.675, TRCoh 39% vs TRInd 22%). A full description of the sensitivities of true MDSE identification among the dose allocation strategies to sample size (n = 102 vs n = 144 vs n = 201), upper asymptote of absolute efficacy (*b*), and prior probability distribution (HS vs HSPL) are reported in the **Supplementary Results in** S1 Appendix (corresponding **S2-S4 Tables in** S1 Appendix). As expected, true MDSE identification improved with increasing sample size, particularly for EA. True MDSE identification was less reliably sensitive to prior probability distribution, with 5–10% differences between HS and HSPL. Decreasing the hypothetical drug's maximum absolute efficacy reduced MDSE identification performance across all dose-response curves, sample sizes, prior distributions, and dose allocation strategies.

### Societal benefit: Probability of individual assignment to true MDSE as proxy for speed

The probability of an individual being assigned to the true MDSE is summarized in **S5 Table in** S1 Appendix. For dose-response curves in which true MDSE was distant from the starting dose level (e.g., curves 1–5), EA and Naïve demonstrated low probabilities of true MDSE assignment (17% and 0%, respectively). TRInd demonstrated a reliably higher rate of true MDSE assignment than TRCoh, ranging from 9% to 38%. Across TRInd, TRCoh, and Gr,

| Allocation Schema | Curve 1 | Curve 2 | Curve 3 | Curve 4 | Curve 5 | Curve 6 | Curve 7 | Curve 8 |
|---|---|---|---|---|---|---|---|---|
| EA | 85% | 41% | 53% | 56% | 64% | 56% | 84% | 84% |
| Naïve | 0% | 0% | 0% | 0% | 0% | 65% | 58% | 94% |
| Gr | 49% | 7% | 26% | 70% | 55% | 77% | 87% | 81% |
| TRInd | 75% | 22% | 49% | 73% | 69% | 73% | 94% | 90% |
| TRCoh* | 78% | 39% | 59% | 66% | 72% | 69% | 92% | 93% |

**Fig 2. Heat map of true MDSE identification probability.** Cells contain the percentage of simulated trials that identified the true MDSE for a given combination of dose allocation strategy and dose-response curve across N = 500 simulated trials, using HS prior distribution, sample size n = 102, and Hill equation parameters corresponding to those in **S1 Table in** S1 Appendix. Darker green color represents a higher probability of this 'good' event; darker red represents a lower probability of this 'good' event. **Abbreviations:** EA, equal allocation; Naïve, naïve allocation; Gr, Greedy allocation; TRInd, targeted randomization (individual); TRCoh, targeted randomization (cohort, comprising 1/3 of study population).

the probability of true MDSE assignment decreased as the number of de-escalation steps "increased" (e.g., TRInd, for curve 7, 74%, to, for curve 2, 28%). A full description of the sensitivities of probability of individual assignment to true MDSE among the dose allocation strategies to sample size (n = 102 vs n = 144 vs n = 201), upper asymptote of absolute efficacy (*b*), and prior probability distribution (HS vs HSPL) are reported in the **Supplementary Results in** S1 Appendix (corresponding **S6-S8 Tables in** S1 Appendix).

## Societal risk: De-escalation from *T* with MDSE underestimation

The probability of each design's estimated MDSE being a lower dose level than true MDSE is summarized in Fig 3. Naïve demonstrated the lowest probability of MDSE underestimation, ranging from 0 to 6% of the simulated clinical trials. Gr returned similarly low probabilities, underestimating MDSE in 0% of simulated clinical trials for 5 out of 8 true dose-response curves; probability of MDSE underestimation in the remaining 3 was 3–19%. TRCoh and TRInd had generally higher rates of MDSE underestimation compared to Naïve and Gr for trials requiring a de-escalation. TRCoh and TRInd had comparable rates of MDSE underestimation to one another: Apart from true dose-response curve 4, in which TRInd had an 19%–8% = 11% lower probability of MDSE underestimation, all other differences were less than 5%. EA underestimated MDSE in 0–37% of cases. In 5 of 8 true dose-response curves, EA's probability of MDSE underestimation was higher than TRCoh; MDSE underestimation occurred at, on average, a 1.7-fold higher rate in EA than TRCoh in these instances. A full description of the sensitivities of MDSE underestimation among the dose allocation strategies to sample size (n = 102 vs n = 144 vs n = 201), upper asymptote of absolute efficacy (*b*), and prior probability distribution (HS vs HSPL) are reported in the **Supplementary Results in** S1 Appendix (corresponding **S9-S11 Tables in** S1 Appendix). The probability of MDSE underestimation was generally insensitive to changes in sample size, but showed moderate sensitivity to prior probability distribution (higher rate of MDSE underestimation with HSPL) and upper asymptote of absolute efficacy (where lower *b* was associated with higher rate of MDSE underestimation) across dose allocation schema.

## Societal risk: Variability of efficacy estimates as assessed by root-mean-squared error (RMSE)

The accuracy of each design's estimated response rate *for its estimated MDSE* using root-mean-squared errors (RMSEs) is summarized in **S12 Table in** S1 Appendix. RMSEs were

| Allocation Schema | Curve 1 | Curve 2 | Curve 3 | Curve 4 | Curve 5 | Curve 6 | Curve 7 | Curve 8 |
|---|---|---|---|---|---|---|---|---|
| EA | 0% | 0% | 2% | 25% | 5% | 37% | 1% | 16% |
| Naïve | 0% | 0% | 0% | 0% | 0% | 0% | 0% | 6% |
| Gr | 0% | 0% | 0% | 3% | 0% | 16% | 0% | 19% |
| TRInd | 0% | 0% | 0% | 8% | 0% | 23% | 0% | 10% |
| TRCoh* | 1% | 0% | 2% | 19% | 3% | 27% | 0% | 7% |

**Fig 3. Heat map of probability of estimating MDSE less than true MDSE.** Cells contain the contain the percentage of simulated trials that estimated an MDSE that is *lower* than the true MDSE for a given combination of dose allocation strategy and dose-response curve across N = 500 simulated trials, using HS prior distribution, sample size n = 102, and Hill equation parameters corresponding to those in **S1 Table in** S1 Appendix. Darker green color represents a lower probability of this 'bad' event; darker red represents a higher probability of this 'bad' event. **Abbreviations:** EA, equal allocation; Naïve, naïve allocation; Gr, Greedy allocation; TRInd, targeted randomization (individual); TRCoh, targeted randomization (cohort, comprising 1/3 of study population).

generally lowest in Gr, Naïve, and TRInd, all designs in which the probability of each individual dose level being MDSE is updated after each observed outcome. RMSE of TRCoh and EA were consistently higher than the others within a given true dose-response curve (e.g., for curve 4, EA 0.069 vs TRCoh 0.056 vs TRInd 0.046). Across all true dose-response curves, $RMSE_{EA} > RMSE_{TRCoh}$. A full description of the sensitivities of RMSE estimates among the dose allocation strategies to sample size (n = 102 vs n = 144 vs n = 201), upper asymptote of absolute efficacy (*b*), and prior probability distribution (HS vs HSPL) are reported in the **Supplementary Results in** S1 Appendix (corresponding **S13-S15 Tables in** S1 Appendix). RMSE estimates were not particularly sensitivity to the choice of prior distribution. Across dose-response curves and dose allocation strategy, RMSE decreased with increasing sample size. There was not a reliable pattern of RMSE change with decreases in upper asymptote of absolute efficacy.

### Individual risk: Probability of underdosing

The probability of an *individual trial subject* receiving a dose less than true MDSE is summarized in Fig 4. Across all dose-response curves, subjects in EA trials had the highest probabilities of being underdosed, offering a consistent baseline for comparison. For all dose-response curves, subjects in Naïve trials had the lowest probabilities of being underdosed. Gr is allowed to assign subjects to a dose level *lower* than the one having highest probability of being MDSE; consequently, subjects in Gr trials tended to have higher probabilities of being underdosed (e.g., for curve 8, Gr 64% vs TRCoh 41% vs TRInd 32%). Within a given allocation strategy, the probability of an individual being underdosed decreases as the estimated MDSE decreases. Of note for the adaptive allocation schema (Gr, Naïve, TRCoh, and TRInd), the true MDSE's proximity to threshold efficacy did *not* reliably increase the probability of underdosing. For example, in dose-response curve 2, where the MDSE is dose level 2 and its efficacy is only slightly above threshold efficacy of 0.6732, the probability of a subject being underdosed ranged from 1% (Gr) to 4% (TRCoh). Both TRCoh and TRInd had lower probability of underdosing than EA across all true dose-response curves. Relative reduction in the probability of a TRCoh subject being underdosed, compared to EA, ranged from 31% (curve 6) to 76% (curve 2). Consistent with less frequent updating of the posterior probability distribution, subjects enrolled in TRCoh trials had generally higher probabilities of being underdosed than subjects enrolled in TRInd trials (mean difference, 2.5%). A full description of the sensitivities of individual subject underdosing among the dose allocation strategies to sample size (n = 102 vs

| Allocation Schema | Curve 1 | Curve 2 | Curve 3 | Curve 4 | Curve 5 | Curve 6 | Curve 7 | Curve 8 |
|---|---|---|---|---|---|---|---|---|
| EA | 17% | 17% | 33% | 50% | 50% | 67% | 67% | 83% |
| Naïve | 0% | 0% | 0% | 0% | 0% | 0% | 0% | 10% |
| Gr | 1% | 1% | 2% | 15% | 5% | 41% | 13% | 64% |
| TRInd | 4% | 3% | 9% | 27% | 15% | 43% | 15% | 32% |
| TRCoh* | 6% | 4% | 10% | 27% | 17% | 46% | 20% | 41% |

**Fig 4. Heat map of probability of administering to an individual subject in the trial a dose less than the true MDSE (underdosing).** Cells contain the average percentage of subjects that were assigned to a dose *lower* than true MDSE for a given combination of dose allocation strategy and dose-response curve across N = 500 simulated trials, using HS prior distribution, sample size n = 102, and Hill equation parameters corresponding to those in **S1 Table in** S1 Appendix. Darker green color represents a lower probability of this 'bad' event; darker red represents a higher probability of this 'bad' event. **Abbreviations:** EA, equal allocation; Naïve, naïve allocation; Gr, Greedy allocation; TRInd, targeted randomization (individual); TRCoh, targeted randomization (cohort, comprising 1/3 of study population).

n = 144 vs n = 201), upper asymptote of absolute efficacy (*b*), and prior probability distribution (HS vs HSPL) are reported in the **Supplementary Results in** S1 Appendix (corresponding **S16-S18 Tables in** S1 Appendix). The dose allocation strategies' probabilities of individual subjects being underdosed were generally insensitive to the prior distribution. The probability of individual underdosing was sensitive to upper asymptote of efficacy in the adaptive designs: As efficacy decreased, the probability of underdosing increased. Other than Gr, which, by design, aggressively assigned to a dose level *lower* than estimated MDSE, the probability of underdosing with the adaptive designs was *lower* than in the classical EA comparator.

## Synthesis

In the context of pandemic-induced scarcity, the major goal behind a dose optimization trial conducted subsequent to a pivotal trial is to generate social value by identifying an MDSE that will mitigate the effect of scarcity. Figs 2 and 3 are the key tables for understanding the social value of de-escalation. No allocation schema is *always* better (dominant) at identifying true MDSE than any other (Fig 2). Several allocation schema are "weakly dominated" in that they are dominated for most curves and only marginally better in the remaining cases. Consider, for example, MDSE identification for TRCoh versus Naïve: TRCoh "defeats" Naïve substantially for all but Curve 8, where TRCoh identifies MDSE 1% less often than Naïve. Naïve can be eliminated by weak dominance. Consider MDSE identification for TRInd versus Greedy. In this case, TRInd dominates for all but Curve 6, where TRInd identifies MDSE at a rate 4% lower than Greedy. Greedy can be eliminated by weak dominance. TRCoh substantially dominates EA in all but Curve 1 and 2, where its performances are 7% and 2% lower, respectively. TRCoh demonstrates comparatively weaker dominance over EA than Naïve but, given uncertainties about which dose-response curve is most likely *a priori*, it is superior. In terms of MDSE identification, neither TRInd nor TRCoh dominates the other; TRCoh identifies MDSE, on average, at a 2.9% higher rate than TRInd. Naïve dominates every other allocation scheme in its likelihood of underestimating MDSE (Fig 3) because it rarely finds the MDSE (Fig 2). TRInd dominates EA. TRInd weakly dominates TRCoh (better in all cases except Curve 8 where it is worse by just 3 percentage points); TRCoh underestimates MDSE at a 2.25% higher rate than TRInd. Targeted Randomization strategies appear to strike the balance between MDSE identification and MDSE underestimation avoidance. Critical for public health emergencies, TRCoh, by dividing the subject population into cohorts, is very likely to proceed at a substantially faster pace than TRInd, allowing for faster identification of socially beneficial results when they exist, in turn increasing social benefits.

## Discussion

In this manuscript, we describe the development of clinical trial methodologies aimed at enabling dose fractionation by de-escalating the dose of a scarce drug previously demonstrated to have efficacy. Such trials must navigate ethical tradeoffs: The extent to which a society is willing to expose individual clinical trial subjects to the risk of an inferior outcome (risk of underdosing) must be balanced against the potential to benefit the population (identification of true MDSE), and the risk of underdosing the population (estimating an MDSE less than true MDSE) must be balanced against a desire to stretch supply of a scarce resource as far as possible (by minimizing the difference between estimated MDSE and true MDSE) [19]. Under these circumstances, adaptive approaches that allow a dose to be de-escalated only up to a point are preferred to a traditional approach that allocates subjects equally across dose levels without regard to observed data (EA). Bayesian dose-escalation studies, which focus on identifying a drug dose that achieves some pre-specified probability of dose-limiting toxicity, are

commonplace in oncologic drug development. However, Bayesian de-escalation approaches, which focus on minimizing total drug exposure, are less prevalent, with the closest analogue being duration optimization trials [33, 34].

Targeted Randomization, in which subjects are assigned to dose levels proportionate to that dose level's estimated probability of being the true MDSE, enables high probability of true MDSE identification, while reducing the risks of underdosing individual trial subjects and underestimating MDSE, as compared to EA. Thus, Targeted Randomization balances dualling objectives: MDSE identification rates were generally equal to or greater than those of EA, Gr, and Naïve, MDSE underestimation rates were lower than EA and comparable to Gr, and underdosing rates were less than EA and comparable to Gr. TRCoh, a cohort-based version of Targeted Randomization, achieves these benefits with infrequent updating of the posterior probability distribution, allowing uninterrupted enrollment of large numbers of subjects. Differences between TRCoh and TRInd tended to be small, with rates of MDSE identification, MDSE underestimation, and underdosing within 5–10% for a given true dose-response curve. Synthesizing the relevant individual and societal tradeoffs as well as the expected cost and time of conducting a de-escalation clinical trial, we conclude that TRCoh is the best of the allocation schemas studied for future application to de-escalation.

Increasing TRCoh trial sample size from n = 102 to n = 201 enabled MDSE identification rate improvement of between 5 and 10%, while other operating characteristics were largely unaffected. Sensitivity analyses showed that TRCoh's advantages in terms of MDSE identification rate, MDSE underestimation rate, and sub-true MDSE dosing were largely preserved with changing sample size. The *number* of subjects exposed to risk of underdosing scales in proportion to sample size, making sample size a value judgment. We cannot make a general recommendation of sample size, whether for TRCoh or any of the other designs evaluated, apart from arguing for the involvement of patients in dose optimization trial development [35]. We expect that a typical sample size determination for these designs would follow a similar logic as for other dose-ranging studies. Specifically, choose an operating characteristic of interest, such the probability of identifying the true MDSE, and then identify the smallest sample size that achieves a certain minimal threshold of that operating characteristic under many different data-generating scenarios. Early stopping rules, such as 'stop if ever the posterior probability that a dose is the MDSE exceeds c', for some prespecified cut point c, could also be incorporated.

HSPL outperformed HS for curves that 1) had lower maximal efficacy and 2) required more de-escalation 'steps' in order to reach true MDSE (e.g., curves 25–27), but HSPL was associated with clearly higher probability of MDSE underestimation. Because de-escalation's greatest risk to populations is MDSE underestimation, utilization of HS is recommended with TRCoh.

The designs presented have relevance beyond pandemic preparedness. Cancer drugs are often administered at higher than minimally necessary doses and more frequently than minimally necessary [18]. This excessive dosing contributes to unnecessary treatment-related toxicity, time burden, financial loss, and environmental damage through unnecessary drug preparation and therapy-related carbon emissions due to travel [18, 36, 37]. Increasing attention is being paid to dose optimization as a means of maximizing clinical benefit, defined as cancer-specific efficacy minus treatment-related toxicities [38]. The presented approach to dose optimization can be employed using a well-established dichotomous endpoint (e.g., objective response rate) and across the range of efficacies observed in cancer treatment (**S4, S8, S11, S15 and S18 Tables in** S1 Appendix) and with less error than conventional Equal Allocation methods over the relevant range of absolute efficacies (**S14 Tables in** S1 Appendix).

There are limitations to this current iteration of de-escalation methodologies and simulation studies. First, our designs are built around a dichotomized outcome measure. Second, our underlying prior distribution enforced monotonicity. Non-monotonic dose-response curves are not uncommon for drugs targeting multiple receptors and drugs for which receptor desensitization, induction of metabolism, and negative feedback exist [39]. *Improved* efficacy with dose de-escalation, as in tuberculosis vaccine H56+IC31 [12], would not stop these trials, the purpose for which is to identify the point within a pre-specified tolerance at which *an individual* would be *no worse off* than if he/she were to receive the default starting dose. Third, our clinical trial simulations evaluate 6 dose levels, an arbitrary number. Previous integrated designs have evaluated $> 10$ evenly-spaced dose levels [40], while previous simulation studies re-examining dose-duration relationships evaluated 7 evenly-spaced intervals. [34] which may not accurately reflect the reality of orally-administered medications. For intravenous medications where multi-dose vials can be used, this issue is much less acute an issue. Fourth, additional improvements will be needed to integrate a stopping rule based on achieving some threshold level of confidence that a given dose level exceeds some threshold level of efficacy. Finally, the "meta-safety" operating characteristic cannot account for *unknowable* downstream events such as resistance to therapy or emergence of vaccine-resistant variants; only time can answer these questions.

Since this research was initiated in 2021, the emergence of monkeypox in the US and UK provided a second instance of potential vaccine shortage. As of 2022, the US had allowed its stockpile of modern smallpox vaccines to decrease to 2,400 doses [41]. Since the smallpox vaccine is also utilized to treat monkeypox, there were insufficient doses available for high-risk populations at the outbreak's onset [41]. In response, the United States and the United Kingdom implemented emergency fractional dosing trials [9, 42]. Fortunately, the monkeypox outbreak subsided. However, it is crucial to be prepared for potential shortages of other vaccines in the future. Conducting dose de-escalation trials in advance can help optimize the use of limited vaccine supplies and maximize their societal benefits.

Dosing of divisible scarce resources is an optimizable choice that may or may not align with a society's broader priorities [1, 2]. Health systems' multiple aims and obligations come into conflict during times of scarcity. The vast majority of societies land between the extremes of "pure equity" and "individual benefit maximization", making reconciliation of interests a necessity. It is important, then, for health systems to understand the *marginal* benefit of a scarce good. When drug development is conducted in the conventional manner (i.e., sequential and non-integrated), the extent to which drug X's benefits depend upon dose, frequency, duration, and route are incompletely known, even after the confirmatory trial–Learning phase trials are imperfect in this respect [43]. De-escalation of the form proposed here is a sensitivity analysis of the effect of drug X's dose on its effect size, the results of which impact how a scarce resource may be utilized. Similar concepts can be leveraged to estimate the effect of modulating *any* factor governed by Hill-type dynamics on efficacy. The de-escalation strategies summarized in this manuscript offer one potential path forward and could be integrated into broader clinical trial platforms and learning health systems to support an two-step model of pandemic drug development in which benefit maximization is pursued immediately after demonstrating efficacy.

## Supporting information

**S1 Appendix.**
(DOCX)

## Author Contributions

**Conceptualization:** Philip S. Boonstra, Garth W. Strohbehn.

**Data curation:** Philip S. Boonstra, Garth W. Strohbehn.

**Formal analysis:** Philip S. Boonstra, Alex Tabarrok, Garth W. Strohbehn.

**Funding acquisition:** Philip S. Boonstra.

**Investigation:** Philip S. Boonstra, Alex Tabarrok, Garth W. Strohbehn.

**Methodology:** Philip S. Boonstra, Alex Tabarrok, Garth W. Strohbehn.

**Project administration:** Garth W. Strohbehn.

**Software:** Philip S. Boonstra.

**Validation:** Philip S. Boonstra, Alex Tabarrok, Garth W. Strohbehn.

**Visualization:** Philip S. Boonstra, Garth W. Strohbehn.

**Writing – original draft:** Garth W. Strohbehn.

**Writing – review & editing:** Philip S. Boonstra, Alex Tabarrok, Garth W. Strohbehn.

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
