## [Decision Letter · Decision Letter 0]

2 Mar 2023

PONE-D-22-21726Targeted randomization dose optimization trials enable fractional dosing of scarce drugsPLOS ONE

Dear Dr. Strohbehn,

Thank you for submitting your manuscript to PLOS ONE. After careful consideration, we feel that it has merit but does not fully meet PLOS ONE’s publication criteria as it currently stands. Therefore, we invite you to submit a revised version of the manuscript that addresses the points raised during the review process.

All comments of both reviewers are essential for you to address prior to publication. 

Please also address the following issues:

(a) In the methods section you state that dose level 0 (placebo) will not be assessed. However, Figure 1 reveals that some of your dose-response curves are up to dose level 4 close to 0.20, the efficacy of dose level 0 (Hill parameter "a"). In other words, whenever one of these corresponding dose levels is chosen, then actually the placebo level is chosen, isn't it?

(b) Figure 1: The Hill equation is a smooth curve, isn't it? You give the impression of a broken lines curve.

(c) Please check and confirm that your R package "DoseDeescalation" meets the criteria of utility, validation, and availability, which are described in detail at http://journals.plos.org/plosone/s/submission-guidelines#loc-methods-software-databases-and-tools.

We look forward to receiving your revised manuscript.

Kind regards,

Harald Heinzl

Academic Editor

PLOS ONE

“I have read the journal's policy and the authors of this manuscript have the following competing interests: PSB has received research funding from Bristol Myers Squibb and Janssen outside of the submitted work. AT has no conflicts to disclose. GWS serves as an uncompensated Director of the Optimal Cancer Care Alliance. GWS is a co-inventor of a patent held by the University of Chicago covering the use of low-dose tocilizumab in the treatment of viral infections. GWS reports no material conflicts of interest with regards to contract research organizations, biostatistical firms, or vaccines.

Disclosures: GWS is employed by the United States Department of Veterans Affairs; this work does not represent the official position of the United States Department of Veterans Affairs.”

Please respond by return email with your amended Competing Interests Statement and we will change the online submission form on your behalf.

Reviewers' comments:

Reviewer's Responses to Questions

**Comments to the Author**

1. Is the manuscript technically sound, and do the data support the conclusions?

Reviewer #1: Yes

Reviewer #2: Yes

2. Has the statistical analysis been performed appropriately and rigorously? 

Reviewer #1: Yes

Reviewer #2: Yes

3. Have the authors made all data underlying the findings in their manuscript fully available?

Reviewer #1: Yes

Reviewer #2: Yes

4. Is the manuscript presented in an intelligible fashion and written in standard English?

Reviewer #1: Yes

Reviewer #2: Yes

5. Review Comments to the Author

Reviewer #1: The authors proposed an innovative Bayesian dose de-escalation design motivated by the vaccine dose-fractioning during the Covid-19 pandemic. The authors conducted thorough simulation studies to evaluate the operational characteristics of various dose assignment algorithms of this design, as well as their sensitivities to the change of several assumptions.

Below are my review comments and questions:

- The abstract claimed that those designs can be seamlessly integrated to platform trials, however this point wasn’t discussed or expanded in the rest of the manuscript. Please clarify what authors mean by the “seamless integration”. Does this mean one seamlessly designed study or two separate studies?

- I am still trying to understand the benefit of conducting dose-de-escalation trials after a drug is demonstrated to be efficacious. In drug development process, dose-selection/dose-finding also occur during phase 1 and phase 2 studies, which usually involves the modeling of the dose response curves and studying both toxicity and efficacy. Is the proposed design only applicable to scenarios where an early-phase dose-finding study to evaluate efficacy has not been conducted?

- Is the sample size of these design pre-specified? If yes, please clarify

o what does the authors mean by the stopping rule (line 149) for dose assignment? Does it just mean reaching the pre-specified sample size?

o Line 211: “speed of a trial depends in part upon the rate at which subjects are assigned to the correct dose level”. I agree there would be benefits for having higher rate of assignment to the correct dose. However, if the sample size pre-specified, then why the rate of getting assigned to the correct dose level would impact the speed of the trial?

o How is the sample size of those studies determined? What recommendations do the authors have for choosing sample sizes?

- What is the computation speed for the proposed design? Does that vary by the dose assignment algorithm?

Reviewer #2: The authors provided a very comprehensive evaluation on the 5 dose randomization schemes for dose fractional trials via numerous simulation scenarios and assumptions.

There are a few areas the author may consider to add clarity in the manuscript

1) Is there any further literature or example (other than COVID) on how the dose fractionation have been adapted in a real trial? I would appreciate if the authors could provide any real trial example in the introduction Section e.g. may be in some disease area that considering dose fractionation trials may be more suitable.

2) From previous literature suggested k =0.7 may be a reasonable relative efficacy target for dose fractionation. I would also recommend the authors may also evaluate both k=0.7 and k=0.8 and to evaluate how results (e.g. probabilities of correctly estimating MDSE, of estimating MDSE less than true MDSE, of underdosing and the error in estimated or true response at the estimated MDSE) may be impacted under all these 5 dose-level randomization schemes.

3) It is unclear what is horseshoe (HS) /positive half of the horseshoe or the horseshoe-plus (HSPI) distribution. Please include more details to describe HS or HSPI.

4) Please provide the rationale on why the four-parameter Hill equation is chosen and is suitable for modelling the dose-response relationship here. Please also consider other parametric dose-response models with fewer parameters (such as the ‘classic’ two-parameter logistic model) for comparisons with the Hill equation. It would be best to include simulation results from another parametric dose-response model (or even considering non-parametric approach for dose-response relationship) for comparison to Hill equation.

6. PLOS authors have the option to publish the peer review history of their article (what does this mean?). If published, this will include your full peer review and any attached files.

Reviewer #1: No

Reviewer #2: No

---

## [Author Response · Author response to Decision Letter 0]

13 Apr 2023

Editor and Reviewer Comments and Authors’ Responses

Editor’s Comments

ED: Please also address the following issues:

(a) In the methods section you state that dose level 0 (placebo) will not be assessed. However, Figure 1 reveals that some of your dose-response curves are up to dose level 4 close to 0.20, the efficacy of dose level 0 (Hill parameter "a"). In other words, whenever one of these corresponding dose levels is chosen, then actually the placebo level is chosen, isn't it?

AU: We appreciate the Editor’s comment. We constructed our “true” dose-response data from dose levels greater than 0. The simulated clinical trial has no knowledge that a dose level 0 exists; instead, it learns that dose level 1 has efficacy of ~0.2. In our construction of the problem, dose level 1 has efficacy nearly equivalent to placebo, but it is not placebo. Extrapolating to a real-world clinical trial, administering placebo when an efficacious alternative exists would be unethical.

ED: (b) Figure 1: The Hill equation is a smooth curve, isn't it? You give the impression of a broken lines curve.

AU: We appreciate the Editor’s attention to detail. The Editor is correct that the Hill equation would produce a smooth curve if continuous-valued inputs were provided. However, we have set up the problem so that the inputs are interpreted as discrete dose levels. That is, for the purposes of simulations, the Hill equation was used to generate true dose-response probabilities for a pre-determined set of discrete, integer values of dose (i.e., 1-6). These probabilities are joined by line segments in Figure 1 only to visually indicate that they belong to the same data generating scenario. After knowing these probabilities, the Hill equation is not further used in the simulated clinical trials. In particular, the statistical model used by the simulated clinical trials to calculate the posterior probability that a given dose level is the minimum dose that achieves the relative efficacy threshold does not rely on the Hill equation in any way. To avoid confusion, we have re-rendered the figure using smooth curves with discrete points. To address this comment and the previous com-ment, we have also modified the discussion about how we used the Hill equation, starting on page 8, line 184, which we believe helps to clear up the meaning of the dose levels and the use of the Hill equation. 

ED: (c) Please check and confirm that your R package "DoseDeescalation" meets the criteria of utility, validation, and availability, which are described in detail at http://journals.plos.org/plosone/s/submission-guidelines#loc-methods-software-databases-and-tools.

AU: We appreciate the Editor’s reminder. We have checked and confirmed that the R package in question adheres to the Journal’s criteria. We would emphasize that this R package is not the primary focus of our contribution but rather provid-ed to readers who are keen to replicate and reproduce our simulation study and potentially make use of our designs, or variants thereof, in their own clinical trials. That said, the DoseDeescalation meets the ‘utility’ criterion by implementing and documenting the designs; it meets the ‘validity’ criterion by providing code for re-creating our simulation study; and it meets the ‘availability’ criterion by being freely available to any R user and documented with a step-by-step vignette on its use. 

 

Reviewer 1

R1: The authors proposed an innovative Bayesian dose de-escalation design motivated by vaccine dose-fractioning dur-ing the Covid-19 pandemic. The authors conducted thorough simulation studies to evaluate the operational characteris-tics of various dose assignment algorithms of this design, as well as their sensitivities to the change of several assump-tions.

The abstract claimed that those designs can be seamlessly integrated to platform trials, however this point wasn’t discussed or expanded in the rest of the manuscript. Please clarify what authors mean by the “seamless integra-tion”. Does this mean one seamlessly designed study or two separate studies?

AU: We appreciate the Reviewer’s close read of the manuscript and for their summary of its main points. We have clari-fied the language in the relevant section of the Abstract and the Discussion.

R1: I am still trying to understand the benefit of conducting dose-de-escalation trials after a drug is demonstrated to be efficacious. In drug development process, dose-selection/dose-finding also occur during phase 1 and phase 2 studies, which usually involves the modeling of the dose response curves and studying both toxicity and efficacy. Is the pro-posed design only applicable to scenarios where an early-phase dose-finding study to evaluate efficacy has not been conducted?

AU: We appreciate the Reviewer’s comment and have added context to the Introduction to make the point more clearly. We agree that, in theory, dose optimization should be done prior to a pivotal efficacy trial, but unfortunately is often not pursued in the real world. This can be due to a “need for speed” (as in COVID-19) or a strong financial incentive to be “first to market” (as in oncology). In practice, the phase 1 and 2 trials conducted in the real world, particularly in both the pandemic and oncology settings, offer insufficient evidence for the phase 3 dose’s optimality. For these situations, dose optimization after pivotal trials is necessary if we are to appropriately distribute scarce resources (in pandemic) or avoid unnecessary toxicity (in oncology).

R1: Is the sample size of these design pre-specified? If yes, please clarify.

AU: We appreciate the Reviewer’s clarifying question. The sample size is not pre-specified in the sense of there existing one globally appropriate choice. Rather, the sample size will generally be chosen subject to statistical, budgetary, and logistical constraints. We have added comments regarding this on page 9 (line 285) and on page 17 (line 494). However, for the purposes of analyzing the sensitivities of the different dose allocation strategies’ performances to the sample size, we considered three different sample sizes (n=102,144,201), representing proportional increases in sample size and an approximate doubling of information from the smallest to the largest sample size. 

R1: What do the authors mean by the stopping rule (line 149) for dose assignment? Does it just mean reaching the pre-specified sample size?

AU: We appreciate the Reviewer’s careful read and have clarified the point in the resubmitted manuscript.

R1: Line 211: “speed of a trial depends in part upon the rate at which subjects are assigned to the correct dose level”. I agree there would be benefits for having higher rate of assignment to the correct dose. However, if the sample size pre-specified, then why is the rate of getting assigned to the correct dose level would impact the speed of the trial?

AU: We appreciate the Reviewer’s careful reading. Ultimately when used in practice, the sample sizes will not be pre-specified, thus the amount of time that passes from the start of the trial to reaching the stopping rule will be dependent upon the rate of assignment to the correct dose level. We have clarified the point in the updated manuscript. In a newly added sentence (line 498, p 17) in response to this comment, we now also note that clinical trialists may want to incorpo-rate an early stopping rule, before the planned maximum sample size, if for example it is very clear from the data that one of the dose levels is the MDSE. 

R1: How is the sample size of those studies determined? What recommendations do the authors have for choosing sample sizes?

AU: We appreciate the Reviewer’s comment. Our primary goal in this manuscript was not to recommend a specific sam-ple size but rather compare the relative performance of these different potential designs given a fixed sample size. That said, there was no specific rationale supporting the sample size for the primary analysis, which was n=102, or 17 subjects per dose level. The sample sizes for the secondary analysis (n=144 and n=201) were chosen as the scale with the sqrt(2), thus allowing us to assess the sensitivity of the trials to doublings of the information present. As each particular trial is idiosyncratic, we would expect a trialist and statistician to begin from a design we have recommended and apply it to the context under study, with careful simulation to guide the final choice. Please see our newly added comments regarding this on page 9 (line 285) and on page 17 (line 494). 

R1: What is the computation speed for the proposed design? Does that vary by the dose assignment algorithm?

AU: We appreciate the Reviewer’s excellent question. In brief, there would be no concerns from a computational perspec-tive in terms of implementing any of our designs, even in the most intensive designs we consider (Naïve, Gr, and TrInd), in which the posterior needs to be updated after every subject in the trial. This is in part due to a clever implementation that we describe in the revised Supplement (Supplemental Methods > Computationally efficient Bayesian update), which obviates the need for a full posterior update each time a subject’s response is recorded. To more specifically answer the reviewer’s valid concern, conducting 10 simulated clinical trials of Gr with 102 subjects, in which the posterior distribution is updated after every subject, i.e. making 1020 Bayesian updates in total, takes about 4.5 minutes on the first author’s personal computer, using the steps in the vignette of the provided R package. In practice, Bayesian updates will be con-ducted one-at-a-time, and so the time or computational cost will be immaterial. We have not included data on computa-tional performance in the updated primary manuscript; we are happy to include a Supplementary table if requested by the Reviewer and Editor. 

Reviewer #2

R2: The authors provided a very comprehensive evaluation on the 5 dose randomization schemes for dose fractional tri-als via numerous simulation scenarios and assumptions. There are a few areas the author may consider to add clarity in the manuscript. Is there any further literature or example (other than COVID) on how the dose fractionation have been adapted in a real trial? I would appreciate if the authors could provide any real trial example in the introduction Section e.g. may be in some disease area that considering dose fractionation trials may be more suitable.

AU: We appreciate the Reviewer’s thoughtful question. We have updated the Introduction and Discussion to include ref-erences to notable past studies of dose fractionation.

R2: From previous literature suggested k=0.7 may be a reasonable relative efficacy target for dose fractionation. I would also recommend the authors may also evaluate both k=0.7 and k=0.8 and to evaluate how results (e.g. probabilities of correctly estimating MDSE, of estimating MDSE less than true MDSE, of underdosing and the error in estimated or true response at the estimated MDSE) may be impacted under all these 5 dose-level randomization schemes.

AU: We appreciate the Reviewer’s particularly thoughtful question. The relative efficacy target for dose fractionation is idiosyncratic to each situation, thus the k used in this study is somewhat arbitrary and not generalizable. The targeted randomization approach concentrates study participants to the dose levels that require the highest resolution. Our da-taset includes difficult “edge cases” where a dose level’s “true” efficacy is just below the threshold level of acceptable efficacy (e.g., curves 4, 6, 12, 13, 20, 22, 28, and 30). Changing k (i.e., moving the dotted line in Figure 1 higher or lower) would not change whether or not these difficult edge cases exist, it would simply change which of the true dose-response curves the edge cases are. On the whole, we would not expect appreciable changes in the operating character-istics, nor would we expect the relative performances of the 5 dose-level randomization schemes to change. 

R2: It is unclear what is horseshoe (HS) /positive half of the horseshoe or the horseshoe-plus (HSPI) distribution. Please include more details to describe HS or HSPI.

AU: We have provided a substantially more detailed description of the HS and HSPl priors in the Supplemental Methods and point to these materials in the Methods section. We elected not to include this in the Methods section directly be-cause we do not feel that these are the main contributions of this manuscript. 

R2: Please provide the rationale on why the four-parameter Hill equation is chosen and is suitable for modelling the dose-response relationship here. Please also consider other parametric dose-response models with fewer parameters (such as the ‘classic’ two-parameter logistic model) for comparisons with the Hill equation. It would be best to include simulation results from another parametric dose-response model (or even considering non-parametric approach for dose-response relationship) for comparison to Hill equation.

AU: We appreciate the Reviewer’s detailed read. We have updated the manuscript (starting on page 8, line 184) to clarify that the dose-response relationship in the simulated clinical trial is not modeled using the Hill equation. Instead, the four-parameter Hill equation was used to develop the “true” dose response curves that the simulated clinical trials were seek-ing to recapitulate. The simulated clinical trial assumes nothing with respect to the nature of the dose-response relation-ship other than monotonicity. The Hill equation was used for the purposes of generating the “true dose-response” da-taset.

---

## [Decision Letter · Decision Letter 1]

7 Jun 2023

Targeted randomization dose optimization trials enable fractional dosing of scarce drugs

PONE-D-22-21726R1

Dear Dr. Strohbehn,

We’re pleased to inform you that your manuscript has been judged scientifically suitable for publication and will be formally accepted for publication once it meets all outstanding technical requirements.

Kind regards,

Harald Heinzl

Academic Editor

PLOS ONE

Additional Editor Comments:

Reviewer 2 has added three important comments, please consider them carefully.

Reviewers' comments:

Reviewer's Responses to Questions

**Comments to the Author**

1. If the authors have adequately addressed your comments raised in a previous round of review and you feel that this manuscript is now acceptable for publication, you may indicate that here to bypass the “Comments to the Author” section, enter your conflict of interest statement in the “Confidential to Editor” section, and submit your "Accept" recommendation.

Reviewer #1: All comments have been addressed

Reviewer #2: All comments have been addressed

2. Is the manuscript technically sound, and do the data support the conclusions?

Reviewer #1: Yes

Reviewer #2: Yes

3. Has the statistical analysis been performed appropriately and rigorously? 

Reviewer #1: Yes

Reviewer #2: Yes

4. Have the authors made all data underlying the findings in their manuscript fully available?

Reviewer #1: Yes

Reviewer #2: Yes

5. Is the manuscript presented in an intelligible fashion and written in standard English?

Reviewer #1: Yes

Reviewer #2: Yes

6. Review Comments to the Author

Reviewer #1: The authors provided detailed and clear responses to all reviewer and editors comments and questions. They also made great additions to the manuscript to clarify those points. I would like to thank the authors for their detailed responses to help me better understand of their work, as well as their efforts to improve the manuscript.

Reviewer #2: Minor Comments:

Thank you for address the review comments and providing updates on the manuscript

Please find the following minor comments:

1) Page 7 lines 171 -172. This sentence discussed about the sum of posterior probabilities equal to 1 and with mean posterior probabilities always equals 1/T could be removed as the same has been discussed earlier in the same page from lines 160 to 161

2) Page 9 lines 204 -205 about the isotonic probability vector distribution. I think these were covered in the supplementary appendix instead of in the main text of this manuscript. Please edit this sentence.

3) Page 9 lines 214 discussing about the choice of K for the relative threshold for efficacy. Please include also your feedback that responded to the reviewer in the manuscript too about k is used ‘depends on situation of each case’, and changes in value of k will only change the ‘edge scenario’ and expected not to change operation characteristics nor the relative performances of the schemes demonstrated in this manuscript.

7. PLOS authors have the option to publish the peer review history of their article (what does this mean?). If published, this will include your full peer review and any attached files.

Reviewer #1: No

Reviewer #2: No

---

## [Editor Report · Acceptance letter]

14 Aug 2023

PONE-D-22-21726R1 

Targeted randomization dose optimization trials enable fractional dosing of scarce drugs 

Dear Dr. Strohbehn:

I'm pleased to inform you that your manuscript has been deemed suitable for publication in PLOS ONE. Congratulations! Your manuscript is now with our production department. 

Kind regards, 

on behalf of

Dr. Harald Heinzl 

Academic Editor

PLOS ONE